# Evaluating the effectiveness of laser hair reduction using a home use laser in comparison to a Diode laser

Kelly Hendricks[1], Celenkosini Thembelenkosini Nxumalo[2], Mokgadi U. Makgobole[1], Shanaz Ghuman[1], Diana Jacobs[3], Nomakhosi Mpofana[1]*

1 Department of Somatology, Durban University of Technology, Durban, South Africa, 2 Academic Development Unit, Faculty of Health Sciences, Durban University of Technology, Durban, South Africa, 3 Somatology Department, Cape Peninsula University of Technology, Cape Town, South Africa

* nomakhosim@dut.ac.za

**Data Availability Statement:** All relevant data are within the paper and its Supporting Information files.

## Abstract

### Background

Lasers of different wavelengths have been developed for use in permanent hair reduction. An increase in the manufacturing of home-use laser hair removal devices allows for these treatments to be performed in the comfort of your own home at an affordable cost.

### Objective

To evaluate the effectiveness of permanent hair reduction using a Diode laser in comparison to the Silk'n™ Flash and Go Lux (475–1200 nm) home-use laser.

### Methods

Fifteen females received six axillae laser hair removal treatments at two to four-week intervals using either a professional laser or home-use laser device. Photographs and hair counts were taken before each treatment and at a three week follow up. A T–test was used to evaluate statistical significance, and regression analysis to determine a difference in the effects. Pain scores and side effects were recorded by a visual analogue scale in a satisfaction questionnaire.

### Results

The professional laser showed an overall hair reduction of 85% on the right axilla and 88% on the left axilla. The home-use laser showed an overall reduction of 52% on the right axilla and 46,3% on the left axilla. Mild side effects were experienced for both laser devices. There were no serious adverse effects reported, safety features are effective to a certain extent.

### Conclusion

The Flash & Go Lux home-use laser can effectively reduce hair at a slower rate than the Diode laser. The home-use laser device offers protection against accidental exposure to

**Funding:** The author(s) received no specific funding for this work.

**Competing interests:** The authors have declared that no competing interests exist.

light and use on darker skin types. Risks of retinal damage due to long-term exposure to home-use laser light are still cause for concern.

## Introduction

Excessive hair growth is a common aesthetic problem noticed in both men and woman. People suffering from abnormal or excessive hair growth are reported to experience psychological, cosmetic and social insecurity [1]. As such, several methods of hair removal have been developed, these include shaving, waxing, tweezing and hair removal creams. These modalities are considered to have temporary results [2] and have largely been replaced by lasers and light sources due to the superiority of these devices in achieving long term hair reduction [3]. The concept of using a light source in the form of a laser for hair removal is known as selective photothermolysis. The mechanism of removal allows for unwanted hair to be selectively removed by targeting the chromophore (melanin) and destroying hair follicles while causing minimal damage to the surrounding tissue [4]. There are three mechanisms of action in which light can destroy hair follicles *i.e.*, Photothermal destruction, photomechanical destruction, and photochemical destruction [3]. Laser technology is based on either high-intensity, coherent, directional, and monochromatic light or intense pulsed light which is based on high-intensity incoherent, polychromatic pulsed light [5]. When light is applied to a biological tissue it can interact with the tissue in four ways; through transmission, reflection, scattering, and absorption using various intensities [6]. According to previous studies, the use of professional lasers such as the Long-pulsed Ruby laser (694 nm), Alexandrite (755 nm), Saprano XL (810 nm), Light sheer Diode (850 nm), Long–pulse neodymium-doped yttrium aluminium garnet (1064 nm) and some sources of Intense Pulsed Light (IPL 500–1200 nm) have been clinically proven and recognised as being the most effective methods of long-term hair reduction [1, 7–9].

The MeDioStar® NeXT Pro Diode laser has a wavelength of 810 nm and 940 nm [10] which allows deeper dermal penetration with less absorption by the epidermal melanin which reduces the risk of adverse side effects in darker Fitzpatrick skin types IV to VI [1]. The ideal candidates for laser and light-based treatment are patients of lighter skin types and dark thick hair [5], however, laser hair reduction can be successfully performed on all Fitzpatrick skin types [4, 11]. Due to the popularity of laser hair reduction (LHR), there has been an increase in IPL manufacturers developing low-powered miniaturized home-use lasers. These systems have been developed to meet the needs of the consumer wishing to undertake permanent hair reduction treatments in the privacy of their home at a more affordable rate than a professional service [12]. There is a range of home-use laser devices including the Silk'n™ Flash & Go with a wavelength of 475–1200 nm [13, 14]. The number of light-based home-use lasers available in the market has significantly increased since the COVID-19 pandemic when social distancing impacted non-essential medical services [15]. According to Austin *et al.* (2021), home-use devices were identified to treat various conditions and concluded that all devices had favourable safety features with few significant side-effects, however, the limitations to the review included a limited amount of randomized controlled trails and a lack of data on the long-term efficacy and safety of the devices [16]. Studies and reviews on home-use lasers also vary in results [17]. Numerous lasers of different wavelengths are available for LHR, however, an increase in various side-effects and complications have been reported when treating Fitzpatrick skin types IV–VI. This is due to the presence of a higher concentration of melanin in the epidermis which functions as a competing absorber of laser light [18]. These side effects and complications can be minimized when the treatment is performed by a professional laser and

skin care therapist [4]. It is therefore important to understand the laser physics, the anatomy of the hair follicle, and skin and tissue response to light in the different skin types to prevent and/or minimise adverse effects from occurring and to ensure a safe and effective treatment.

The study aimed to establish the effectiveness of laser hair reduction using two different laser device systems in terms of outcomes, side effects and adverse effects.

## Methods and materials

### Clinical data

Thirty female participants were recruited between 15 April and 30 June 2022 from an existing database and word-of-mouth. The participants had no previous laser treatments done on the axilla area. The participants were aged between 20 to 30 years with a mean age of 26.11 and 28.17 for each test group. The participants were categorised to be within the Fitzpatrick skin types III and IV, and all treated hair was noted to be dark terminal hairs. Participants who indicated interest were screened to ensure that they met the inclusion criteria. The screening included an initial telephonic survey or video call followed by a thorough examination which included taking a case history in the form of a record card. The inclusion criteria which qualified female individuals to participate included participants with Fitzpatrick skin type III–IV, females between the age of 20–30, participants with unwanted hair in the axilla area, first-time laser participants, participants who have not had previous laser treatments on the axilla area before, participants who were not suntanned, participants who were not contraindicated to laser, willing, committed and agree to the pre and post-care procedures required of the treatment. Females were selected to assess one gender and avoid multiple variables such as hormonal differences between genders. The age range was chosen due to being a period before menopause. At the age of 20, the body has gotten past the hormonal inconsistencies of adolescence and has established a rhythm. At the age of 30, the body begins to experience slight hormonal changes and at 35 these changes increase [19].

Once the study sample had been established, each participant was assigned with a reference number which was generated randomly to assign each member to a particular laser machine. Exclusion criteria utilised in this research included the following: participants with blonde, red or light-coloured hair under their arms, sunbathers and tanned participants, pregnant and breastfeeding participants, subjects suffering from epilepsy and seizures, individuals having received previous professional or home laser treatments on the axilla area, skin diseases and disorders, bacterial, fungal or viral infections, persons prone to hypertrophic scarring or keloids, patients on certain medication such as Isotretinoin, antibiotics, anticoagulants, aspirin, tattoos in the area, those experiencing joint pain–under gold therapy treatment, contagious disease, diabetics, suspicious pigmented lesions in the area being treated, use of photosensitive drugs, and use of phototoxic drugs.

### Research design

This is a quantitative, experimental research study. Participants were randomly assigned into two groups A and B. A number generator was used to assign the participants to receive hair removal treatments either with the MeDioStar ®NeXT Pro Diode laser or the Silk'n™ Flash & Go home laser.

### Study setting

The study was conducted at Deluxe laser Clinic Constantia in the Western Cape, Cape Town from 15 July 2022 to 30 December 2022. The study setting took place in a designated laser

room which complied with the standard guidelines and requirements of a laser room (S1 Appendix). It was also labelled as a controlled area with a warning sign fixed on the outside of the door. Protective eyewear was used for both the home-use laser and the professional laser. The home-use laser machine's make and model were covered with white adhesive tape in order protect the manufactures identity.

## Ethical approval and considerations

The study methods and protocols were approved by the IREC (IREC 095/22) at the Durban University of Technology. A written informed consent was obtained prior to each subject's participation and no minors were included in the study. Each participant was identified through their reference number by authors and only the main author and supervisors had access to information that could identify participants during and/or after data collection. The researcher attended the Research Integrity seminar which focused on the Introduction to research ethics. Gatekeeper permission was sought from Deluxe Laser & Spa's store manager to conduct research on the premises.

## Data collection process

Thirty female participants received six sessions of laser hair reduction from 15 July 2022 to 30 December 2022. Written informed consent was obtained and the risk, benefits and nature of the study was discussed. Baseline measurements were taken before the treatment by using an electronic structured standardized questionnaire in the form of a record card. The record card included a separate section for the researcher to record settings and findings. All participants were required to refrain from any methods of hair removal for two to four weeks and allow their axilla hair to grow before their first treatment. Prior to the treatment, initial hair counts were recorded within a 2 $cm^2$ which was marked using a body pencil under both underarms. Hair counts and photographs were taken before each treatment. A shaving medium was applied to the axilla area and the hair was shaved flush to the skin. The area was cleansed and sanitized using a non-scented wet wipe. The cooling gel was applied to the area being treated and protective goggles were placed over the participants eyes and researcher's eyes. The treatment parameters were adjusted according to the participants' skin type. For the professional laser, a spot size of 24x38 mm was used and a test pulse was performed on 6 or 8 $J/cm^2$. If no erythema or blistering was noted after five minutes the fluence was increased to one setting higher. If any side effects were noted, the fluence was reduced until a suitable setting was established. The treatment was then administered on the entire area competing two passes vertically and horizontally overlapping by 0.5 cm. If the participant experienced any excessive heat on the treatment site, the area was immediately cooled with the sapphire cooling headpiece, and additional cooling post-laser. After the laser, the gel was then removed from the area, cleaned using a damp cloth and an aloe vera cream was applied. Treatments with the professional laser were performed four weeks apart with a follow up of three weeks after the final session.

With the home use-laser, the cooling gel was removed, and the area was cleaned and dried prior to the treatment. The laser parameters were adjusted according to the manufacturer's test pulse recommendations of 1 or 2 $J/cm^2$. If no adverse effects were noted after five minutes, energy levels were increased to 3 or 4 $J/cm^2$ depending on the skin type. Eventually all participants were treated with the highest energy levels of 4.5–5 $J/cm^2$. The entire area was treated by completing two passes horizontally and vertically, with an overlap of 0.5cm. If the participant experienced any excessive heat during the treatment while the shots were being administered, some cooling gel was applied to that area for a few minutes. The cooling gel was then reapplied to the entire treated area for post-cooling, it was then removed after a few minutes and cleaned

with a damp cloth. Soothing aloe vera cream was then applied to the area. Treatments were performed bi-weekly for sessions one to four and sessions five and six were spaced four weeks apart according to the manufacture's recommendations with a follow-up of three weeks. During the course of the treatments, participants were instructed to avoid all other methods of hair removal between treatments besides shaving. Participants were allowed to shave the area two weeks after the laser session if the treatment was four weeks apart (device dependent). It was further stipulated that participants not shave the area after the laser session if the treatment was two weeks apart (device dependent). Post-care and treatment instructions were provided to each participant and were provided with guidelines for shaving to establish uniformity. During the follow up appointment, a final hair count was performed, the treated area was assessed for any adverse effects and a satisfaction questionnaire was provided to each participant for feedback on their personal experience and observations. The questionnaire included a visual analogue scale and pain analogue scale to measure participants' pain, redness and discomfort. Photographs and a hair count were taken before each treatment, after six treatments and three weeks post-completion of the study period during a follow-up visit. A final follow-up appointment was scheduled three weeks after the completion of the treatment period to take final measurements.

### Data analysis

All the collected data were recorded and transferred onto the latest Statistical Package for Social Sciences (SPSS) software package for statistical analysis. Various descriptive and inferential statistical techniques was used. Tables, and summary statistics *i.e*, means, proportions, and percentages were used as descriptive measures to identify trends in the data collected. During the course of statistical analysis, data were analysed using a parametric or nonparametric test normality of the variable based on the form of the variables. All statistical analyses were completed at the 95% level of significance ($\propto$ = 0.05). If the *p*–value, as reported is less than 0.05, a statistically significant result was declared.

## Results

A total of thirty female participants were enrolled for the research of which fifteen completed the study sessions. All the data were analysed using a parametric or nonparametric test normality of the variable based on the form of the variables. All statistical analyses were completed at the 95% level of significance ($\propto$ = 0.05). The statistical significance was considered to be *p*<0.05. Based on hair density recorded for the Diode laser at the beginning and after six sessions, the overall reduction of hair on the right axilla was estimated to be 85% and 88% on the left axilla. The hair density recorded for the home-use laser at the beginning and after six sessions showed an overall reduction of 52% on the right axilla and 46,3% on the left axilla.

### Hair reduction

The average hair count with the Diode laser on the right axilla decreased significantly from pre-treatment with a mean = 76.22 to post-treatment mean = 11.56, t (8) = 11.177. The left axilla decreased significantly from pre-treatment with a mean = 76.56 to post-treatment mean = 9.56, t (8) = 11.753, $p < .001$. There was a statistical significance of $p < .001$ for both the right and left axilla.

The average hair count with the home-use laser on the right axilla decreased significantly from pre-treatment with a mean = 97.83 to post-treatment with a mean count = 47.00, t (5) = 4.691. The statistical significance measured for the right axilla was $p = 0.05$. The left axilla

**Table 1. Regression analysis for group type (type of laser) on the right axilla.**

| IV | $R^2$ | F | df1; df2 | *p*-value | B (regression coefficient) | t | *p*-value |
|---|---|---|---|---|---|---|---|
| PreRight | .556 | 7.518 | 2; 12 | .008 | .404 | 1.490 | .162 |
| Group | | | | | -26.703 | -2.320 | .039 |

DV = Post Right

decreased significantly from pre-treatment with a mean count = 89.50 to post-treatment with a mean count = 48.00, t (5) = 4.006, and statistical significance of *p* = .010.

To determine the differences in the effect of the two laser treatments the ANCOVA test was performed and reported as follows:

Table 1 analyses shows that when the post-scores are 'standardized' using pre-scores the group (type of laser used) significantly affects post-scores. Post-hair counts were significantly more when using the home laser (estimated M = 41.76) than when using the professional laser (estimated M = 15.05, *p* = .039.).

Table 2 analyses showed that skin type marginally affected post hair counts (*p* = .080) with skin type III resulting in a higher post-count (estimated M = 36.71) than skin type IV (estimated M = 18.68).

Table 3 analyses shows that post hair counts are significantly more when using the home laser (estimated M = 44.44) that when using the professional laser (estimated M = 11.93), *p* = .009.

Table 4 analyses shows that when the post-scores are 'standardized' using pre-scores and further corrected for with skin type, the group (type of laser used) significantly affects post-scores. Post-hair counts are significantly more when using the home laser (estimated M = 43.33) that when using the professional laser (estimated M = 11.72), *p* = .008.

The results of visual assessment for participants before and after six sessions for the Diode laser are shown in Figs 1–4 shows the visual assessment for the home-use laser. Label A1 and A2 represents all the before photos, and label B1 and B2 represents all after photos. Hair counts within a 2 cm$^2$ area were obtained before each treatment as well as the follow up appointment.

## Side effects

The side effects assessment during the treatment is shown in Table 5. Mild side effects were experienced by some participants for both laser types. Approximately, 66.7% of participants undergoing professional laser treatment experienced pain ranging from mild to moderate, 33.3% experienced slight redness or mild erythema and 22.2% experienced mild inflammation. All side effects subsided within minutes post-treatment.

A total of 66.7% of participants experienced pain from the home-use laser. A further 16.7% of the participants experienced redness and 16.7% experienced inflammation which only presented at the time of the treatment.

**Table 2. Regression analysis for group and skin type on the right axilla.**

| IV | $R^2$ | F | df1; df2 | *p*-value | B (regression coefficient) | t | *p*-value |
|---|---|---|---|---|---|---|---|
| PreRight | .668 | 7.386 | 3; 11 | .006 | .270 | 1.058 | .313 |
| Skin | | | | | 18.036 | 1.928 | .080 |
| Group | | | | | -25.607 | -2.460 | .032 |

DV = PostRight

**Table 3. Regression analysis for group (type of laser used) on the left axilla.**

| IV | R² | F | df1; df2 | p-value | B (regression coefficient) | t | p-value |
|---|---|---|---|---|---|---|---|
| PreLeft | .598 | 8.942 | 2; 12 | .004 | .458 | 1.653 | .124 |
| Group | | | | | -32.517 | -3.095 | .009 |

DV = PostLeft

Although there were guidelines for shaving frequency, 78% of participants being treated with the professional laser never shaved between sessions as they never felt the need to shave whereas 22% shaved once between sessions. With the home-use laser, 50% of participants did not shave between sessions, 16.67% shaved twice between sessions, 16.67% shaved three times and 16.67% shaved four times.

## Discussion

Previous studies and reviews on professional lasers showed positive results in the reduction of hair [5]. Fifteen of the thirty participants completed all six sessions including a three week follow up. The results show a positive reduction of terminal hairs for the professional laser with an initial mean hair count under the right axilla of 76.22 which was reduced to 11.56 after six sessions. The left axilla had an initial mean hair count of 76.56 which reduced to 9.56 after six sessions. The results for the home-use laser observed a positive reduction with an initial mean hair count of 97.83 under the right axilla which reduced to 47.00. The left axilla had an initial mean hair count of 89.50 and reduced to 48.00 after six sessions.

Comparing the effects of the two lasers, the post-counts for the home-use laser was greater than the post-counts for the professional laser. This translates to the professional laser promoting a greater reduction in hair than the home-use laser. Skin type and post-hair counts were also evaluated, which resulted in an observably greater post-hair count in Fitzpatrick skin type III than Fitzpatrick skin type IV. Meaning that Fitzpatrick skin type IV had a greater reduction in hair than Fitzpatrick skin type III. This could be due to a greater amount of melanin in the hair shaft, follicle, and bulb in Fitzpatrick skin type IV.

When receiving laser hair reduction treatments there are side effects associate with it ranging from mild and expected to long-term complications [20]. A review conducted by Arsiwala and Majid (2019) reported that the effectiveness of laser is not only dependant on the laser device but on the person performing the treatment and significant adverse effects have been associated with laser being performed by untrained physicians [3].

According to the evaluation of the side effects experienced in this study, participants experienced a certain degree of pain although ranging from mild to moderate, transient redness and inflammation were not reported as severe cases. All side effects subsided within minutes post-treatment which was due to following the correct professional laser protocols, which includes selecting the correct treatment parameter settings with sufficient epidermal cooling before, during and after the laser treatment.

**Table 4. Regression analysis for group and skin type on left axilla.**

| IV | R² | F | df1; df2 | p-value | B (regression coefficient) | t | p-value |
|---|---|---|---|---|---|---|---|
| PreLeft | .676 | 7.636 | 3; 11 | .005 | .235 | .798 | .442 |
| Skin | | | | | 17.083 | 1.618 | .134 |
| Group | | | | | -31.608 | -3.199 | .008 |

DV = PostLeft

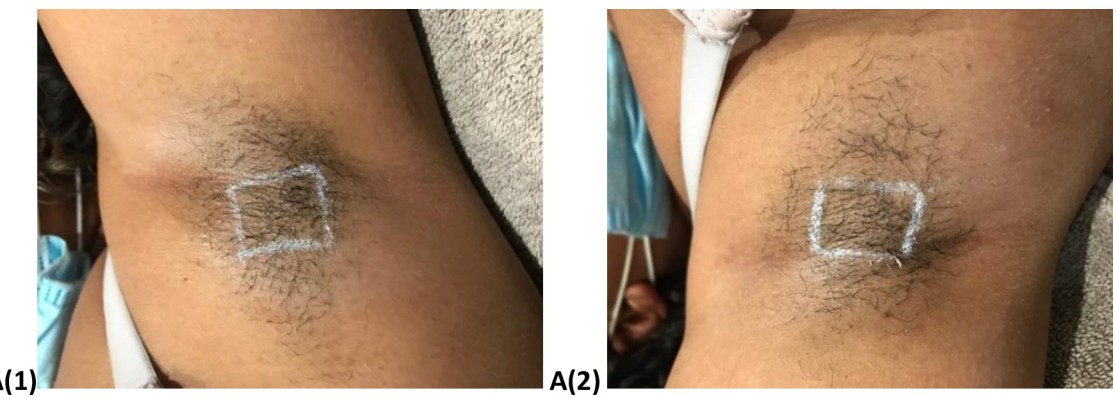

**Fig 1.** Before images of left (A1) and right (A2) axilla.

A report on IPL home devices stated that the devices are not as safe as initially reported and when using professional laser systems, there is a thin line between achieving the appropriate outcome and the manifestation of complications [17]. A review conducted by Austin *et al.* (2021) on home-based devices reported that home-pulsed IPL devices are regarded as a safe therapeutic option for laser hair reduction, however, darker skinned patients should use low-fluence IPL. It was also reported that reviewed studies do not provide enough evidence regarding the recommended treatment parameters [16].

Professional laser hair removal treatments have various complications which could be minimized when the treatment is performed by a well-trained skin care therapist [21]. When it comes to home-use laser devices, there are contradictory results where safety is concerned. While performing the laser treatments in this study, from general observations, the home laser device does not include any built-in cooling mechanism for epidermal cooling even though the light is able to generate heat and cause some pain or discomfort. A fair amount of heat can be felt even on the lowest fluence setting without any transient erythema occurring. Goggles are not provided due to manufactures' claiming that the light is contained within the device, however, while performing the treatment the light tends to escape on the sides of the headpiece similar to the professional Diode laser and is visible even while wearing protective broad-spectrum goggles. Elm *et al.* (2010) reported that this light is not harmful to the eye [22]. The most effective safety feature is the built-in colour sensor which is designed to measure the

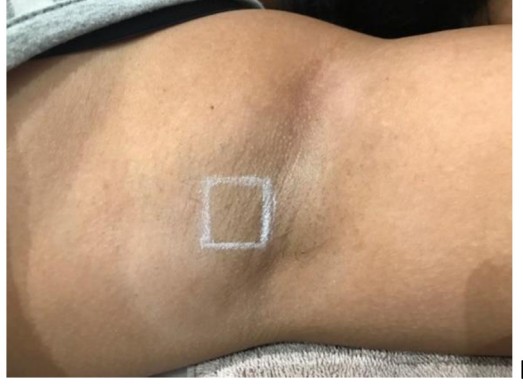
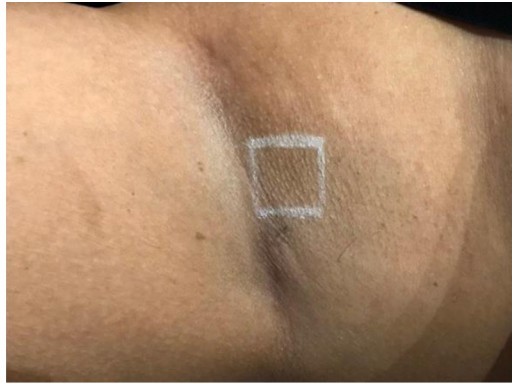

**Fig 2.** After images of left (B1) and right (B2) axilla.

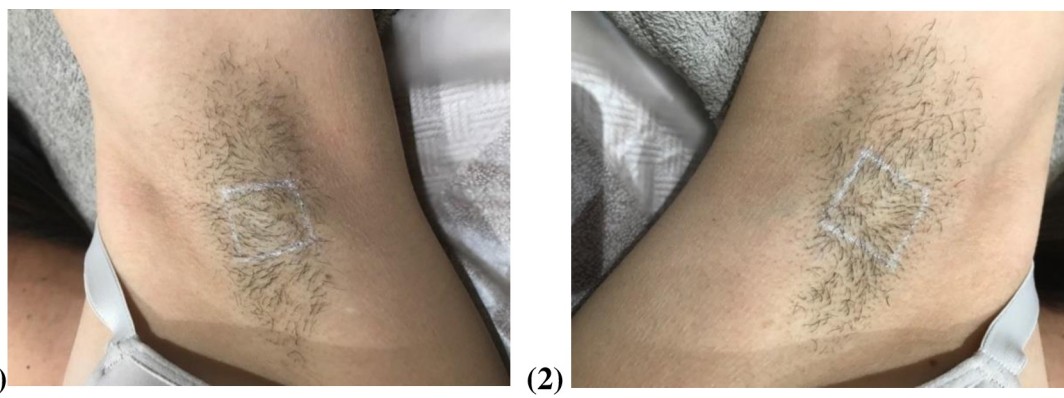

**Fig 3.** Before images of left (A1) and right (A2) axilla.

complexion of the surface on which the laser is applied. This enables the application of the laser to only work on suitable skin complexions and will not emit pulses if placed on a darker skin type. The laser device has been designed to only emit pulses when the spot size is in full contact with the skin tissue and cannot emit any pulses while facing open air. Studies by Alster and Tanzi (2009) and Elm *et al.* (2010) reported that the Home Skinovations Silk'n ™ Flash & Go should not be used on Fitzpatrick skin types V–VI and not used on facial, neck, tanned or sun exposed skin [13, 22]. Contradicting the studies is the manufacturer's guide stating that the Silk'n™ Flash & Go can be used to treat facial hairs with extra caution to avoid the eye area [14].

The Mediostar has been approved by the US Food and Drug Administration (FDA) and considered safe and effective for clinical use. The Silk'n ™Flash & Go does not state on any packaging or packaging content that the device is FDA-approved, however, a study on the Flash & Go stated that the device had been approved by the FDA in 2008. The FDA website verifies a Silk'n™ laser however, it is not for the Flash & Go device [23]. The Flash & Go is verified by a simple google search for the 510K number of the Silk'n™ Flash & Go which provides a PDF document of its FDA approval.

## Conclusions

Previous research and studies concluded that laser and light sources have become one of the most popular procedure in dermatology and skin care in treating unwanted hair. Due to the

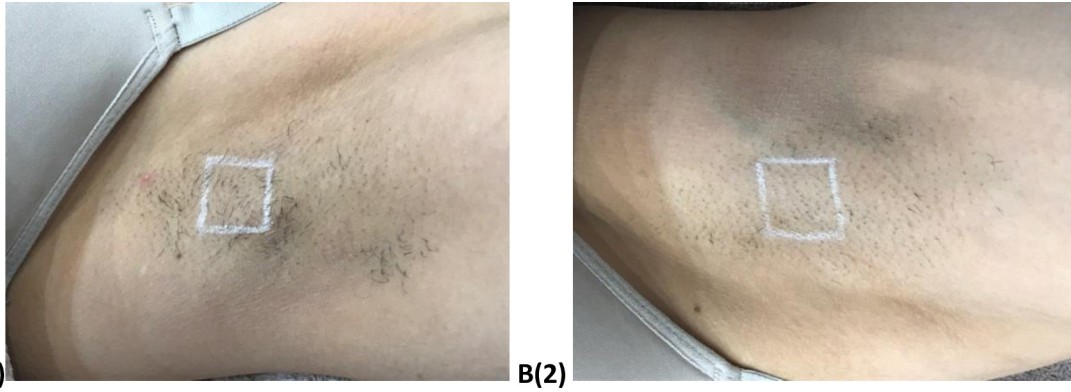

**Fig 4.** After images of left (B1) and right (B2) axilla.

**Table 5. Percentage of side effects according to laser type.**

| Side effects | Professional laser | | Home Laser | |
|---|---|---|---|---|
| | N | Percentage | N | Percentage |
| Pain | 6 | 66.7 | 4 | 66.7 |
| Redness | 3 | 33.3 | 1 | 16.7 |
| Inflammation | 2 | 22.2 | 1 | 16.7 |

popularity of laser hair reduction, many IPL manufactures developed miniature home-use laser devices to administer treatments in the comfort of their homes. Whether laser hair reduction is performed in a professional setup by a well-trained skin care therapist or at home by inexperienced members of the public, possible side effects and adverse effects can occur. Knowledge on how these effects can be treated is important especially when using a home use laser whereas, these effects are reduced when laser is being performed by a knowledgeable therapist. The home-use laser device used in this study has integrated safety mechanisms such as skin colour sensors to help prevent thermal burns and skin contact sensors to reduce open air light exposure in aid of preventing possible retinal injury, however, the user of the device may still be at risk despite these safety mechanisms. Both methods of hair reduction showed positive reductions from the pre-hair count to the post-hair count. The difference in effects between the two laser devices showed a greater reduction in hair counts in participants treated with the Diode laser. Changes in hair texture were also greater in participants treated with the Diode laser although, there were textural hair changes in participants being treated with the Flash & Go laser device as well. The difference in skin type also affected the hair count with a higher post-hair count in skin types III. Side effects were experienced between both laser devices none of which were serious or long term, and no adverse effects were reported.

The concern for retinal damage due to long term exposure still remains a safety concern regardless of the integrated safety mechanisms. Whether the light is contained within the home laser device, safety goggles should be part of the packaging content even if there is no guarantee that the user will wear them.

Although home-based laser devices have received approval by the FDA, it is evident that lasers and home-use laser devices are not completely harmless and should not be used at home without the initial support and supervision of a medical or skin therapist.

## Supporting information

**S1 Appendix. Basic requirements of a laser room.**
(DOCX)

## Acknowledgments

Authors would like to express their gratitude to pen2paperedits for editing the manuscript.

## Author Contributions

**Conceptualization:** Kelly Hendricks.

**Formal analysis:** Mokgadi U. Makgobole, Diana Jacobs.

**Investigation:** Kelly Hendricks, Diana Jacobs.

**Methodology:** Kelly Hendricks, Celenkosini Thembelenkosini Nxumalo, Nomakhosi Mpofana.

**Software:** Mokgadi U. Makgobole, Nomakhosi Mpofana.

**Supervision:** Shanaz Ghuman, Nomakhosi Mpofana.

**Writing – original draft:** Kelly Hendricks.

**Writing – review & editing:** Celenkosini Thembelenkosini Nxumalo, Shanaz Ghuman, Diana Jacobs, Nomakhosi Mpofana.

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
