## [Decision Letter · Decision Letter 0]

10 Apr 2023

PONE-D-23-07622Evaluating the Effectiveness of Laser Hair reduction using a Home use Laser in comparison to a Diode LaserPLOS ONE

Dear Dr. Mpofana,

Thank you for submitting your manuscript to PLOS ONE. After careful consideration, we feel that it has merit but does not fully meet PLOS ONE’s publication criteria as it currently stands. Therefore, we invite you to submit a revised version of the manuscript that addresses the points raised during the review process.

We look forward to receiving your revised manuscript.

Kind regards,

Michael R Hamblin

Academic Editor

PLOS ONE

Journal Requirements:

Additional Editor Comments:

You must be clear about the difference between laser and IPL. The mechanisms of action need to be further explained. The limitations must be further discussed. Additional references are required as mentioned

Reviewers' comments:

Reviewer's Responses to Questions

**Comments to the Author**

1. Is the manuscript technically sound, and do the data support the conclusions?

Reviewer #1: Yes

2. Has the statistical analysis been performed appropriately and rigorously? 

Reviewer #1: I Don't Know

3. Have the authors made all data underlying the findings in their manuscript fully available?

Reviewer #1: Yes

4. Is the manuscript presented in an intelligible fashion and written in standard English?

Reviewer #1: Yes

5. Review Comments to the Author

Reviewer #1: The title is inaccurate and misleading. It should read: "Evaluating the effectiveness of laser hair reduction using a home-use intense pulsed light device in comparison with a professional diode laser"

There must be a distinction between 'laser' and 'intense pulsed light' as they are radically different technologies and this must be maintained throughout the article to avoid misunderstanding.

The mode of action in the introduction needs to be clarified and expanded.

Retinal injury is not the only risk, consider the pigmented iris!

In the discussion, there is insufficient reference to published data on safety of home-use IPL devices (see attachment).

Te statements about loss of skin contact and light spillage require clarification and expansion (see attachment).

The text about US FDA 510k pre-marketing clearance of the devices used in this study relates only to permission to sell a product in the USA based on predicate information and has no relevance to this study. Reference FDA clearances is correctly used in the conclusions.

The Conclusions need to be revised and expanded to include limitations of this study e.g. small sample size, need for more randomised, controlled studies, etc .(see attachment).

Several important references are missing including recent publications in international peer-reviewed journals including:

Trelles M, Ash C, Town G. Clinical and Microscopic Evaluation of Long-Term Epilation Effects of the iPulse Personal Home-Use Intense Pulsed Light (IPL) Device. J Eur Acad Dermatol Venereol. 2014; 28(2):160-168.

Town G, Botchkareva N, Uzunbajakava N, Nuijs A, van Vlimmeren M, Ash C, Dierickx C. Light-based home-use devices for hair removal: Why do they work and how effective they are? Lasers Surg Med. 2019; 51:481-490. Early View online: https://onlinelibrary.wiley.com/doi/10.1002/lsm.23061.

Hatterskey AM, Kiernan M, Goldberg D, Dierickx C, Sliney DH, Haedersdal M. Nash JF. Assessment of adverse events for a home-use intense pulsed light hair removal device using postmarketing surveillance. Lasers Surg Med 2023; Mar 8. doi: 10.1002/lsm.23650.Online ahead of print.

Town G, Ash C, Dierickx C, Fritz K, Bjerring P, Haedersdal M. Guidelines on the safety of light‐based home‐use hair removal devices from the European Society for Laser Dermatology. J Eur

Acad Dermatol Venereol. 2012;26(7):799–811. https://doi.org/10. 1111/j.1468-3083.2011.04406.x

6. PLOS authors have the option to publish the peer review history of their article (what does this mean?). If published, this will include your full peer review and any attached files.

Reviewer #1: **Yes: **Godfrey Arthur Town Ph.D.

---

## [Author Response · Author response to Decision Letter 0]

2 May 2023

We would like to express our heartfelt appreciation to the reviewers for their invaluable contributions. All of the changes suggested by the reviewers have been implemented. The manuscript now reads much better.

---

## [Editor Report · Decision Letter 1]

3 May 2023

PONE-D-23-07622R1Evaluating the effectiveness of laser hair reduction using a home-use intense pulsed light device in comparison with a professional diode laserPLOS ONE

Dear Dr. Mpofana,

Thank you for submitting your manuscript to PLOS ONE. After careful consideration, we feel that it has merit but does not fully meet PLOS ONE’s publication criteria as it currently stands. Therefore, we invite you to submit a revised version of the manuscript that addresses the points raised during the review process.

 You have changed the abstract in the submission system but have not changed the text in the paper. Perhaps you submitted the wrong version? You must replace all instances of "home use laser" with "home use IPL device" Please submit a version with all the changes highlighted

We look forward to receiving your revised manuscript.

Kind regards,

Michael R Hamblin

Academic Editor

PLOS ONE

---

## [Author Response · Author response to Decision Letter 1]

9 May 2023

We have changed the text in the paper as advised. All changes made are underlined in the "revised manuscript".

---

## [Editor Report · Decision Letter 2]

10 May 2023

Evaluating the effectiveness of laser hair reduction using a home-use intense pulsed light device in comparison with a professional diode laser

PONE-D-23-07622R2

Dear Dr. Mpofana,

We’re pleased to inform you that your manuscript has been judged scientifically suitable for publication and will be formally accepted for publication once it meets all outstanding technical requirements.

Kind regards,

Michael R Hamblin

Academic Editor

PLOS ONE
---

## [Editor Report · Acceptance letter]

18 May 2023

PONE-D-23-07622R2 

Evaluating the effectiveness of Laser hair reduction using a Home use Laser in comparison to a Diode Laser 

Dear Dr. Mpofana:

I'm pleased to inform you that your manuscript has been deemed suitable for publication in PLOS ONE. Congratulations! Your manuscript is now with our production department. 

Kind regards, 

on behalf of

Dr. Michael R Hamblin 

Academic Editor

PLOS ONE